# Five-year clinical outcomes of 107 consecutive DMEK surgeries

**Pierre Bichet[1], Rémi Moskwa[1], Christophe Goetz[2], Yinka Zevering[2], Jean-Charles Vermion[1], Jean-Marc Perone[1]** *

**1** Ophthalmology Department, Mercy Hospital, Regional Hospital Center (CHR) of Metz-Thionville, Metz, France, **2** Clinical Research Support Unit, Mercy Hospital, Regional Hospital Center (CHR) of Metz-Thionville Regional Hospital Center, Metz, France

* jean-marc.perone@chr-metz-thionville.fr

**Data Availability Statement:** The datasets generated during and/or analyzed during the current study are not publicly available according to French Law No. 2018-493 of June 20, 2018 on the protection of personal data (The General Data

## Abstract

### Purpose

The long-term clinical outcomes, postoperative complications, and graft survival of Descemet-membrane endothelial keratoplasty (DMEK) remain poorly understood. We retrospectively assessed these variables in all consecutive eyes that underwent DMEK for any indication in 2014–2018. The findings were compared to the long-term DMEK studies of five other groups (3–10-year follow-up).

### Methods

Patients underwent ophthalmological tests preoperatively, at 1, 3, 6, and 12 postoperative months, and then annually. Five-year graft survival was determined by Kaplan-Meier estimator. Change in best-corrected visual acuity (BCVA), endothelial-cell density (ECD), and central-corneal thickness (CCT) at each timepoint was determined.

### Results

107 eyes (80 patients; 72 years old; 67% female) underwent first-time DMEK for uncomplicated Fuchs endothelial corneal dystrophy (94% of eyes), pseudophakic bullous keratopathy (3%), and regraft after previous keratoplasty (3%). The most common complication was graft detachment requiring rebubbling (18%). Thirteen grafts (12%) failed at $\leq$15 months. Cumulative 5-year graft-survival probability was 88% (95% confidence intervals = 79–94%). BCVA improved from 0.6 logMAR preoperatively to 0.05 logMAR at 1 year (p<0.0001) and then remained stable. Donor ECD dropped by 47% at 6 postoperative months and then continued to decrease by 4.0%/year. Five-year endothelial-cell loss was 65% (from 2550 to 900 cells/mm$^2$). CCT dropped from 618 to 551 μm at 5 years (p<0.0001). These findings are generally consistent with previous long-term DMEK studies.

Protection Regulation (Regulation (EU) 2016/679) (GDPR: article 9) but are available from the Clinical Research Support Platform (Plateforme d'Appui à la Recherche Clinique [PARC]) of the Regional Central Hospital (CHR) of Metz-Thionville on reasonable request (email: projet-recherche-clinique@chr-metz-thionville.fr, tel: +33 3 87 17 98 82). All non-archived data is subject to daily backups while all archived data is subject to duplicate storage at two different sites. This data processing is compliant with a baseline reference methodology (MR-001) to which the CHR Metz-Thionville signed a compliance commitment on October 8, 2018.

**Funding:** The authors received no specific funding for this work.

**Competing interests:** The authors have declared that no competing interests exist.

## Conclusions

DMEK has low complication and high graft-survival rates and excellent clinical outcomes that persist up to 5 years post-surgery. DMEK seems to be a safe and effective treatment in the long term.

## Introduction

In 2006, a ground-breaking keratoplasty technique emerged, namely, Descemet-membrane endothelial keratoplasty (DMEK). This posterior lamellar keratoplasty technique, which was invented by Melles [1,2], involves transplanting a 15 μm-thick graft consisting of Descemet membrane and corneal endothelium only. It is particularly suitable for treating Fuchs endothelial corneal dystrophy (FECD), which is the most common corneal dystrophy [3]. This progressive inherited condition is characterized by pleomorphic, attenuated, and degenerated corneal endothelium and presents clinically as reduced vision due to corneal edema [4]. The edma reflects the growing inability of the endothelium to pump the fluid out of the corneal stroma [5]. Another common indication for DMEK is pseudophakic bullous keratopathy (PBK), which is an iatrogenic condition caused by damage to the endothelial-cell layer during phacoemulsification treatment of cataracts [6]. Other indications include previous graft failure, trauma, infection, and uveitis [7]. Prior to the development of DMEK, these corneal-endothelial pathologies were treated with penetrating keratoplasty, which involves transplanting a full-thickness (540–560-μm) corneal graft [8]. However, the outcomes were often poor due to high rates of immune rejection, astigmatism, and slow and inadequate visual recovery [9–11]. These outcomes were improved by Descemet-stripping automated endothelial keratoplasty (DSAEK), which is a lamellar keratoplasty technique that was invented by Melles a few years before DMEK [12]. However, DSAEK involves a thicker graft that bears some stroma. As a result, DMEK results in much faster and better visual recovery with less immune rejection. These attributes have dramatically changed the keratoplasty field, with FECD becoming the leading indication for endothelial keratoplasty and DMEK increasingly being used for FECD and uncomplicated PBK [7,9,10,13].

However, DMEK is a technically challenging procedure because of the thinness of the graft and its tendency to scroll tightly; this also significantly increases the risk of graft detachment [14,15]. This has slowed the global uptake of DMEK by corneal surgeons.

Given the relatively short duration since DMEK was first proposed and tested in the field, nearly all studies on DMEK only report short-term (6–24 month) outcomes. To our knowledge, five groups have reported DMEK outcomes beyond 2 years (summarized in Table 1) [16–23]. While they suggest that the complication rate is low, graft survival is high, and long-term clinical outcomes are excellent, some of these studies are limited by the fact that they excluded indications and/or eyes with vision-limiting comorbidities and thus do not reflect real-world outcomes. Moreover, the other studies, which were from the Melles and Kruse groups, examined the earliest cases in the field (2007–2012) and thus may not fully reflect the clinical practices that evolved after the initial pioneering work. In addition, most studies included second eyes in their statistical analyses (Table 1), which may violate assumptions of subject independence. Loss to follow-up was also high. Thus, we report the outcomes of all consecutive eyes that underwent first-time DMEK in 2014–2018 in our institution. The cohort was not selected in any way. All eyes were examined. Separate analyses were also conducted with first eyes only, and first FECD eyes only. Unlike most other long-term follow-up studies (Table 1), the vast majority of the patients were available at the 5-year follow-up.

**Table 1. Long-term outcomes of DMEK in consecutive case series in the literature.**

| First author date[ref] Study period | No. eyes (pts) | Indic. | Postop time point | Eyes lost to FU/GF | Patient age, y | Female sex | Donor age, y | Triple DMEK | Preop BCVA, logMAR[a] | Preop graft ECD, c/mm2[a] | Preop CCT, µm[a] | Rebub or major graft detach | CME & rejection | At last follow-up unless otherwise indicated | | | Graft survival prob | %PGF & SGF (timept SGF)[b] |
|---|---|---|---|---|---|---|---|---|---|---|---|---|---|---|---|---|---|---|
| | | | | | | | | | | | | | | BCVA[a] (% ≥ decimal) | ECD, c/mm2[a] (% ECL) | CCT, µm[a] (% loss) | | |
| Our study 2014–2018 | 107 (80) | Mixed (94% FECD, 3% PBK; 3% failed KP) | 5y | 13% | 72 | 67% | 77 | 32% | 0.6 | 2550 | 618 | 18% | 6% | 0.0 | 900 (65%) | 550 (11%) | 88% | 6.5% & 5.6% (6m) |
| Zwingelberg 2022[23] 2015–2016 | 402 (402) | Mixed (92% FECD, 8% PBK) | 3y | NR | 74 FECD 73 PBK | 67% FECD 40% PBK | 67 FECD 70 PBK | NR | 0.44 FECD 0.88 PBK | 2710 FECD 2728 PBK | 682 FECD 932 PBK | 30% FECD 28% PBK | NR | 0.18 FECD 0.20 PBK | 1594 FECD (41%) 1542 PBK (44%) | 527 FECD (23%) 664 PBK (29%) | NR | NR |
| Weller 2022 [35] (Kruse group) 2009–2012 | 66 | Mixed (91% FECD, 9% failed KP) | 8y 9y 10y | 0% 59% 32% | 63 | 53% | 69 | 42% | 0.63 | 2582 | 650 | 45% | 9% & 0% | 0.19 0.18 0.13 | 739 (71%) 744 (71%) 729 (72%) | 559 (12%) 563 (13%) 563 (13%) | NR | NR & 6% (>8y) |
| Besek 2022 [21] 2013–2019 | 150 (137) | Mixed (41% FECD, 59% PBK) | 6m 1y 3y 5y 7y | 1% 3% 45% 75% 87% | 66 | 56% | 53 | 27% | 1.62 | 2572 | 665 | 13% | NR & 5% | 0.73 0.51 0.38 0.43 0.38 | 1793 (37%) 1688 (43%) 1532 (45%) 1310 (48%) 1190 (55%) | 568 (9%) 572 (7%) 572 (10%) 575 (13%) 578 (10%) | 58% 7y | 11% & 9.3% (>1y) |

Included: Consecutive ≥18-y-old patients who underwent DMEK ≥5 years previously. Excluded: first five patients.

Included: Consecutive 18–95-y-old patients with FECD or PBK with 3y FU after uncomplicated DMEK. Excluded: extracorneal visual acuity–limiting conditions (AMD, diabetic retinopathy, macular edema, advanced glaucoma, amblyopia); previous surgery (vitrectomy, pressure-reducing glaucoma surgery, and previous corneal surgery); patients with insufficient data.

Included: Consecutive eyes with 8y minimum FU (66 of total 450 eyes = 85% LTFU). For the BCVA data, eyes with comorbidities that could limit visual outcome (11% of eyes; amblyopia, epithelial basement membrane dystrophy, optic nerve atrophy) were left in (data above) or excluded. When comorbid eyes were excluded, preoperative BCVA = 0.55 logMAR, 8y postop BCVA = 0.10 logMAR.

Included: Consecutive patients who underwent DMEK for FECD or PBK between 0.5 and 8 years previously. Excluded: conditions that might affect vision (retinal detachment, recurrent uveitis, trauma, proliferative diabetic retinopathy, macular holes, glaucoma, retinal vascular occlusion, AMD); <6 months FU. No exclusions for three surgeons' learning curves.

(Continued)

**Table 1.** (Continued)

| First author date[ref] Study period | No. eyes (pts) | Indic. | Postop time point | Eyes lost to FU/GF | Patient age, y | Female sex | Donor age, y | Triple DMEK | Preop BCVA, logMAR[a] | Preop graft ECD, c/mm2[a] | Preop CCT, μm[a] | Rebub or major graft detach | CME & rejection | At last follow-up unless otherwise indicated | | | Graft survival prob | %PGF & SGF (timept SGF)[b] |
|---|---|---|---|---|---|---|---|---|---|---|---|---|---|---|---|---|---|---|
| | | | | | | | | | | | | | | BCVA[a] (% ≥ decimal) | ECD, c/mm2[a] (% ECL) | CCT, μm[a] (% loss) | | |
| Wardeh 2020 [20] 2010–2014 | 230 (142) | Mixed (94% FECD, 6% PBK) | Mean 47 (range 20–80) m | 42% since surgery | 69 | 60% | NR | 0.8% | 0.60 | 2559 | 675 | 19.6% | 2.6% & 1% | 0.10 (71% ≥0.01; 96% ≥0.5) | 1166 (55%) | 547 (19%) | 92% 6.3 y | 4.8% & 3.0% (several m-4y) |
| Included: Consecutive patients who underwent DMEK in 2010–2014 and attended FU at 20–80 mo. 125 of 265 patients (165/395 eyes) did not attend = LTFU of 47% (42%). For the BCVA data, eyes with low visual potential (AMD, macular pucker, amblyopia, choroidal neovascularization in myopia, optic nerve atrophy) were excluded. | | | | | | | | | | | | | | | | | | |
| Vasiliauskaite 2020[19] (Melles group) 2007–2009 | 100 (88) | Mixed (94% FECD, 2% PBK, 4% failed KP) | 5y | 32% | 68[8] | 59% | 62 | 16% | 0.36 | 2593 | 668 | 11% | NR & 4% | 0.04 (98% ≥0.5) | 1083 (59%) | 540 (19%) | 83% | 0% & 6% (mean 5y) |
| | | | 10y | 43% | | | | | | | | | | 0.03 (98% ≥0.5) | 845 (68%) | 553 (17%) | 79% | |
| Included: First 100 consecutive DMEK cases (cases 26–125). Excluded: first 25 cases; for the BCVA analysis, eyes with low visual potential due to ocular co-morbidities unrelated to the cornea (≥8% at all timepoints; for survival analysis, second operated eyes (n = 12). First cases of DMEK in the world. | | | | | | | | | | | | | | | | | | |
| Birbal 2020 [18] (Melles group) 2007–2012 | 500 (393) | Mixed (89% FECD, 6% PBK, 3% failed KP, 1% other) | 5y | 28% | 68 | 54% | 65 | 25% | 0.49 | 2530 | 667 | 8.8% | NR & 3% | 0.05 (99% ≥0.5) | 1140 (55%) | 539 (19%) | 90% 5y | 0.2% & 2.8% |
| Included: All consecutive eyes. Excluded: first 25 cases; for BCVA analysis, eyes with low visual potential due to ocular comorbidities unrelated to the cornea (≥12%); for BCVA, patients not presenting for FU; for survival analysis, second eyes (n = 107). First cases of DMEK in the world. | | | | | | | | | | | | | | | | | | |
| Ham 2016[17] (Melles group) 2007–2009 | 250 (209) | Mixed (89% FECD, 7% PBK, 4% failed KP) | 7y | 87% | 67 | 56% | NR | 20% | NS (35% >0.5–0.8, 65% >1.0) | 2553 | 670 | 4.4% | NR & 2.4% | NS (96% ≥0.5) | 1210 (53%) | 536 (20%) | 96% both 4y & 7y | 2% & 2% (>12m) |
| Included: All consecutive eyes, including 25 learning curve eyes. Excluded: For BCVA analysis, cases with low visual potential due to concomitant ocular pathology (retina or optic nerve disorders) (≥8%). First cases of DMEK in the world. | | | | | | | | | | | | | | | | | | |

*(Continued)*

**Table 1.** (Continued)

| First author date[ref] Study period | No. eyes (pts) | Indic. | Postop time point | Eyes lost to FU/GF | Patient age, y | Female sex | Donor age, y | Triple DMEK | Preop BCVA, logMAR[a] | Preop graft ECD, c/mm2[a] | Preop CCT, μm[a] | Rebub or major graft detach | CME & rejection | At last follow-up unless otherwise indicated | | | Graft survival prob | %PGF & SGF (timept SGF)[b] |
|---|---|---|---|---|---|---|---|---|---|---|---|---|---|---|---|---|---|---|
| | | | | | | | | | | | | | | BCVA[a] (% ≥ decimal) | ECD, c/mm2[a] (% ECL) | CCT, μm[a] (% loss) | | |
| Schlogl 2016 [16] (Kruse group) 2009–2011 | 97 (84) | Mixed (91% FECD, 1% PBK, 7% failed KP, 1% other) | 3y | 23% | 66 | 51% | 67 | 51% | 0.62 | 2602 | 644 | 10.8% | 3% & 1% | 0.13 (97% ≥0.5) | 1460 (44% ECL) | 557 (13%) | NR | 2% & 2% (23/39m) |
| | | | 5y | 59% | | | | | | | | | | | | | 95% | |

Included: All consecutive eyes with a minimum of 3 years follow-up. Excluded: graft failures with secondary keratoplasty (n = 12); for BCVA analysis, eyes with ocular comorbidities limiting visual outcome; for survival analysis, the initial learning curve of the surgeons involved (operations conducted within first 6 months after introduction of DMEK at center).

[a] BCVA, ECD, and CCT data are expressed as mean in the literature but as median in our study because these variables were generally not normally distributed. If we had expressed these variables as means, preoperative BCVA (without 7 eyes with ocular comorbidities), ECD, and CCT would be 0.7 logMAR, 2574 cells/mm², and 633 μm, respectively; and 5-year postoperative BCVA, ECD, and CCT would be 0.1 logMAR, 900 cells/mm², and 561 μm, respectively.

[b] PGF = absence of corneal clearance after surgery; SGF = new corneal edema due to corneal decompensation.

AMD, age-related macular degeneration; BCVA, best-corrected visual acuity; CME, cystoid macular edema; CCT, central corneal thickness; detach, detachment; DMEK, Descemet endothelial keratoplasty; ECD, endothelial cell density; FECD, Fuchs endothelial corneal dystrophy; FU, follow-up; GF, graft failure; indic, indication; KP, keratoplasty; LTFU, loss to follow-up; m, months; NR, not recorded; PBK, pseudophakic bullous keratopathy; PGF, primary graft failure; pts, patients; rebub, rebubbling; ref, reference number; SGF, secondary graft failure; timept, timepoint; y, year.

## Materials and methods

### Study design and ethics

This retrospective single-center cohort study was conducted in the Metz-Thionville Regional Hospital Center (Grand Est, France). It was performed in accordance with the principles of the Helsinki Declaration and was approved by the Ethics Committee of the French Society of Ophthalmology (Approval No. 00008855). All patients consented in writing to the surgery and the potential use of their medical data after being informed about the risks and benefits of the procedure. The consent procedure was conducted in accordance with the reference methodology MR-004 of the National Commission for Information Technology and Liberties of France (No. 588909).

### Patient selection

The prospectively collected medical records were searched for all consecutive adult ($\geq 18$ years) patients whose eye(s) underwent first-time pseudophakic-DMEK or triple-DMEK (cataract surgery followed by DMEK) between March 2014 and March 2018 in the Department of Ophthalmology of the Metz-Thionville Regional Hospital Center. The first five eyes that underwent DMEK in our series were excluded to limit learning-curve effects.

### Graft preparation, surgical techniques, postoperative treatment, and follow-up

All patients underwent preoperative lower peripheral iridotomy with Nd-YAG laser (Laser ex-Super Q; Ellex Europe, Medical Quantel, Cournon d'Auvergne, France) to prevent preoperative and postoperative pupillary block. All surgeries were performed by the same experienced surgeon (JMP). General anesthesia was used in all but two cases: these patients underwent locoregional (peribulbar) anesthesia due to general anesthesia contraindications.

DMEK was performed as described by Melles et al. [24]. All grafts were from two French regional tissue banks (Besançon or Nancy). They were stored in organ-culture medium (Eurobio) at 31°C until transplantation and had a requested endothelial-cell density (ECD) of >2000 cells/mm$^2$. First, the graft was prepared in the operating room by creating an 8 mm-diameter disc with a Hanna microtrephine (Busin Punch 17200D 8mm single use; Moria SA, Antony, France). The Descemet membrane and endothelium were then manually stripped off the corneal stroma with a disposable curved monofilament forceps (Single Use Tying Forceps Curved 5mm Platform 17501; Moria SA, Antony, France) under a microscope. Subsequently, the endothelial graft was colored with Trypan Blue (Vision Blue, 0.5-mL syringe; D.O.R.C. Dutch Ophthalmic Research Center, Zuidland, Netherlands), marked on its stromal side with an E or F, and placed in a customized injector (30G Curved cannula for air injection; D.O.R.C. Dutch Ophthalmic Research Center, Zuidland, Netherlands).

The recipient cornea was then prepared. Thus, an 8-mm circular marker that had been inked with a dermographic pen was pressed to the central surface of the cornea. The main paracentesis was then placed supero-temporally (for the right eye) or supero-nasally (for the left eye) with a 2.2-mm blade (Securityblade BD, Xstar 2.2-mm, 45 degrees, 37822; Beaver-Visitec International, CityPoint Waltham, USA). A second incision was made with a Worst 15 blade (ophthalmic knife 15 degrees; ALCON, Rueil Malmaison, France). A 9-mm central descemetorhexis was conducted along the circular mark with an inverted Sinskey Price hook (Single Use Price Reverse Hook Sim 17302; Moria SA, Antony, France) and an inverted spatula (90th single use Spatula 17303; Moria SA, Antony, France) under sterile air infusion. The main incision was then enlarged to 4 mm with the 2.2-mm blade and the graft was injected into the

anterior chamber with the D.O.R.C. injector. The main incision was sutured with one point of Nylon 10.0 that was secondarily buried. The graft was unfolded by external corneal pressure generated by two 27-gauge Rycroft cannulas. When the graft was well positioned, a sterile air or 20% sulfur hexafluoride bubble was injected into the anterior chamber to keep the graft in place.

If the patient was phakic at the time of DMEK, DMEK was preceded by phacoemulsification in all cases (triple-DMEK). The phacoemulsification procedure was a standard subluxation technique [25]. It was conducted with Stellaris PC (Bausch and Lomb, Aliso Viejo, CA, USA). A Zeiss CT Asphina 409MV intraocular lens was implanted in the capsular bag. The postoperative refractive target was a residual myopia of -0.5 to -1.00 diopters to compensate for the hypermetropizing effect of DMEK surgery [26]. MIOSTAT 0.01% (Carbachol; ALCON, Rueil Malmaison, France) was injected to induce miosis and facilitate the DMEK that followed.

After surgery, all patients were treated for 4 weeks with Maxidrol, which is a topical antibio-corticosteroid (Dexamethasone + Neomycin Polymyxine B; ALCON, Rueil Malmaison, France; four times/day), and an ophthalmic ointment (Vitamin A dulcis; ALLERGAN, Courbevoie, France; twice/day). Maxidrol was then tapered for a month and replaced with long-term low-dose corticosteroid eye drops (FLUCON; Novartis Pharma, Rueil Malmaison, France; three times/day).

Follow-up with complete ophthalmological testing was conducted 1, 8, and 15 days and 1, 3, 6, and 12 months after surgery and every year thereafter. Postoperative cystoid-macular edema (CME) was diagnosed by posterior-segment optical-coherence tomography (OCT) (RS-3000; Nidek Co. Ltd, Japan). If visual-acuity loss was observed, it was treated with oral acetazolamide (Diamox®; Sanofi, Gentilly, France; one 250 mg tablet 3 times/day for 1 month) and NSAID (indomethacin 0.1%; Chauvin, Montpellier, France; 4 times/day for one month). Allograft rejection was defined as a line of retro-descemetic precipitates and was treated for 1 month with an anti-inflammatory corticosteroid ointment (Sterdex®, which contains dexamethasone with oxytetracycline; Thea, Clermont-Ferrand, France; twice/day) and Maxidrol (ALCON, Rueil Malmaison, France; 12 times/day for 1 week then 8, 6, and 4 times/day for the second, third, and fourth week, respectively). When graft detachment was clinically suspected, anterior-segment OCT (RS-3000; Nidek Co. Ltd, Japan) was conducted. If more than a third of the graft surface area and/or the visual axis was threatened, rebubbling (i.e. injection of a sterile air bubble in the anterior chamber) was conducted immediately under topical anesthesia. It was repeated up to four times. Primary-graft failure (PGF) was defined as the failure of the pre-existing edema to resolve within 3 months of surgery, as measured with anterior segment-optical coherence tomography [27] at follow-up visits. Secondary-graft failure (SGF) was defined as the new emergence of corneal edema after 3 months. All PGF and SGF cases were treated with regraft with DMEK or DSAEK.

## Preoperative and postoperative testing

Patients were examined before surgery, at 1, 3, 6, and 12 postoperative months, and then annually for 5 years after DMEK. At each time point, the patients underwent a full ophthalmic examination that consisted of: slit lamp examination; measurement of best-corrected visual acuity (BCVA) using the Monoyer chart, after which the values were expressed as minimum angle of resolution logMAR for statistical analysis; measurement of central-corneal thickness (CCT) with non-contact ultrasonic pachymetry (Tono pachymeter NT-530P; Nidek Co., Gamagori Aichi, Japan); and anterior-segment OCT (NIDEK with a special module; Nidek Co., Gamagori Aichi, Japan). The preoperative ECD of the graft was measured by the eye

bank. The postoperative ECD was assessed with a specular microscope (CEM-530; Nidek Co., Ltd., Gamagori, Aichi, Japan). Postoperative ECD values at 1 and 3 months and some CCT values at various timepoints were missing due to postoperative edema, which made these measurements unreliable.

### Preoperative, perioperative, and postoperative variables

The following data were obtained from the prospectively maintained clinical database: patient age, gender, graft indication, and operated eye side; graft donor age and ECD; use of triple-DMEK; type of anesthesia; and preoperative and postoperative BCVA, ECD, and CCT. Graft-donor age and ECD were provided by the eye bank.

### Statistical analyses

Continuous data were expressed as median (interquartile range [IQR]) while categorical variables were expressed as $n$ (%). All patients were included in the longitudinal analysis until they were lost to follow-up or underwent regraft. Graft-survival probability was determined by Kaplan-Meier analysis. The data were expressed as percent survival at 5 years with 95% confidence intervals (CIs). The survival analysis was conducted with the whole cohort (first plus second eyes) but since inclusion of second eyes could violate statistical assumptions of subject independence, Kaplan-Meier analysis was also repeated with the first eyes only. Moreover, to focus on the primary indication for DMEK [7], the survival analysis was repeated with the first eyes with FECD. Postoperative change in the clinical variables relative to preoperative values was assessed for statistical significance with Wilcoxon signed rank test. To be consistent with the statistical practices in the other long-term DMEK studies [16–23], these analyses were conducted with the whole-eye cohort. However, since Wilcoxon test assumes subject independence, the analyses were repeated with first eyes only, as well as with first FECD eyes only. For all BCVA analyses, the eyes that had preoperative comorbidities that could affect vision were excluded. This also aligned with the practices in the previous studies [16–23]. Missing ECD and CCT values are indicated in the tables and were ignored during analysis. P-values of <0.05 were considered to indicate statistical significance. All statistical analyses were performed with SAS software (version 9.4, SAS Inst., Cary, NC, USA).

## Results

We started conducting DMEK in 2014. Between 2014 and 2018, 112 eyes from 85 patients underwent first-time DMEK. Of these, the first five were excluded because they represent the learning curve of this new technique. Consequently, 107 eyes from 80 patients were included in the study (Fig 1). Thus, in 25% of patients, DMEK was performed bilaterally, meaning 27 of the eyes in the cohort were second eyes.

### Preoperative patient and eye characteristics

The 80 patients were on average 72 years old and 67% were women. Of the 107 eyes, half were right eyes, 32% underwent triple-DMEK, mean graft-donor age was 77 years, and FECD was by far the most common indication (94%). The mean preoperative BCVA of the eyes was 0.6 logMAR: these eyes (n = 100) did not include the seven eyes that had preoperative ocular comorbidities that could affect visual acuity (neovascular age-related macular degeneration [nAMD] n = 4; retinal epimacular membrane n = 2; glistening of the lens implant after prior cataract surgery n = 1). The mean preoperative CCT was 618 μm (these data were only

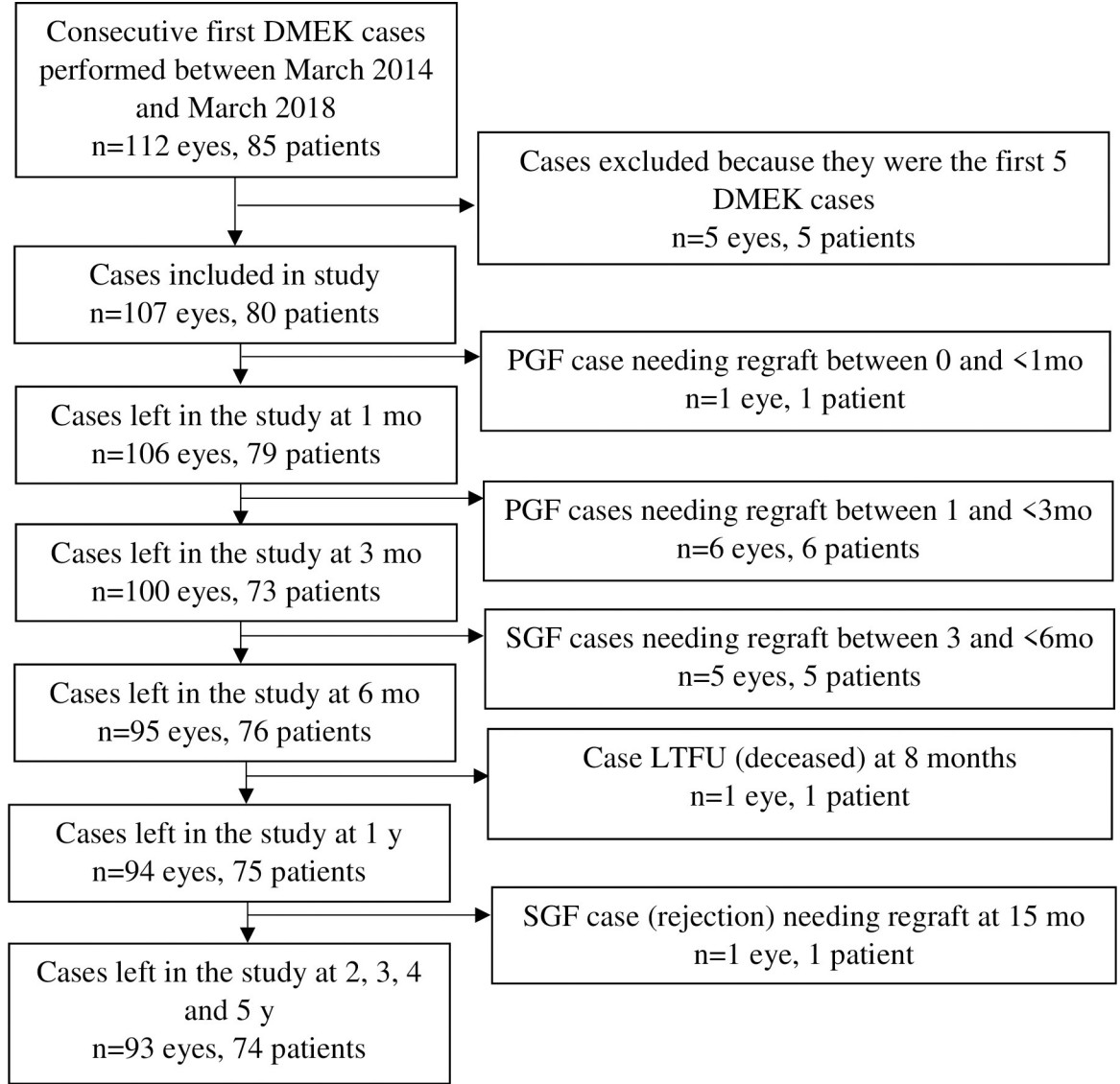

**Fig 1. Patient distribution during the 5-year study period.** LTFU, loss to follow-up; mo, months; PGF, primary graft failure (≤3 months); SGF, secondary graft failure (>3 months); y, years.

available for 97 eyes). The mean graft ECD was 2550 cells/mm$^2$. The first- and second-eye sub-cohorts did not differ markedly in terms of these variables (Table 2).

## Complications after DMEK surgery and graft survival

The main complication was graft detachment: of the 107 eyes, 20 (18%) went under at least one rebubbling procedure for major detachment. Rebubbling was successful in 14 cases (70%). Six of the grafts that did not undergo rebubbling failed due to excessive deterioration of the endothelium that caused poor recovery of visual acuity. There was also one allograft rejection case at 15 months. Thus, in total, 13 eyes required retransplantation with DSAEK (n = 12) or a second DMEK (n = 1). All regrafts were successful. Of these 13 failure cases, seven were PGF cases: one failed before 1 month and six failed between 1 and <3 months (overall median, 2 months). The remaining six were SGF cases: five failed between 3 and <6 months and one

**Table 2. Patient demographics and DMEK characteristics.**

| Preoperative Variable | Total cohort | First eyes | Second eyes |
|---|---|---|---|
| No. eyes/patients | 107/80 | 80/80 | 27/27 |
| Patient age, years | 72 (66–78) | 74 (66–79) | 71 (66–77) |
| Female sex | 71 (67) | 53 (66) | 18 (67) |
| Right eye | 55 (51) | 42 (53) | 13 (48) |
| Lens status | | | |
| Phakic [i.e. triple DMEK] | 34 (32) | 24 (30) | 10 (37) |
| Pseudophakic [i.e. DMEK alone] | 73 (68) | 56 (70) | 17 (63) |
| Donor age, years | 77 (71–82) | 77 (70–82) | 77 (73–84) |
| Indication for DMEK | | | |
| FECD | 101 (94) | 74 (93) | 27 (100) |
| PBK | 3 (3) | 3 (4) | 0 (0) |
| Failed PKP/DSAEK graft | 3 (3) | 3 (4) | 0 (0) |
| Other | 0 (0) | 0 (0) | 0 (0) |
| BCVA, logMAR (n = 100)† | 0.6 (0.5–0.7) | 0.7 (0.5–1) | 0.5 (0.4–0.7) |
| Graft ECD, cells/mm$^2$ | 2550 (2433–2717) | 2560 (2427–2725) | 2520 (2433–2650) |
| CCT, μm (n = 97) | 618 (580–647) | 620 (580–649) | 616 (556–640) |

Data are number (percentage) or median (interquartile range [IQR]). The sample size for each variable was 107 unless otherwise indicated.

† Eyes with preoperative conditions that could affect visual function recovery were excluded from the BCVA analysis. BCVA, best-corrected visual acuity; CCT, central corneal thickness: DMEK, Descemet membrane endothelial keratoplasty; DSAEK, Descemet stripping automated endothelial keratoplasty; ECD, endothelial cell density; FECD, Fuchs endoethlial corneal dystrophy; PBK, pseudophakic bullous keratopathy; PKP, penetrating keratoplasty.

failed at 15 months (overall median, 5 months). One eye was lost at 8 months due to the death of the patient. No eyes were lost between 2 and 5 years. Thus, of the 107 eyes, 14 (13%) were lost by the 5-year follow-up. All were first eyes, meaning that 14 of the 80 patients (18%) were lost before 5 years (Fig 1).

There were six cases (6%) of postoperative CME. All were treated with oral acetazolamide and topical NSAID. Four resolved completely while the remaining two retained some minimal edema. In addition, two eyes developed non-graft-related comorbidities after DMEK surgery, namely, cilioretinal-artery occlusion at 11 months and nAMD at 3 years. Neither associated with graft loss.

The Kaplan-Meier analysis of the whole cohort (first and second eyes) showed that graft survival at 5 years was 88% (95% CIs = 79%–94%) (Fig 2). The survival of the first eyes only was also 88% (95%CIs = 77–95%), while the survival of the first FECD eyes only was 89% (95% CIs = 78–95%) (S1 Fig).

## BCVA, ECD, and CCT outcomes over time

To be consistent with the statistical practices of the other long-term DMEK studies [16–23], the total-eye cohort was assessed for changes in BCVA, ECD, and CCT over the study period. However, identical patterns were observed for the first eyes only (S1 Table). This was also true for the first eyes of the FECD patients only (S2 Table). The seven eyes with vision-limiting pre-operative comorbidities were excluded from the BCVA analyses. Thus, median (IQR) BCVA in the whole cohort improved steadily from 0.6 (0.5–0.7) to 0.05 (0–0.1) logMAR at year 1, after which it dropped further to 0 (0–0.1) logMAR at 2 years and stabilized (Fig 3A). At 5 years, BCVA had decreased by 0.6 logMAR to 0 (0–0.15) logMAR. The changes at all

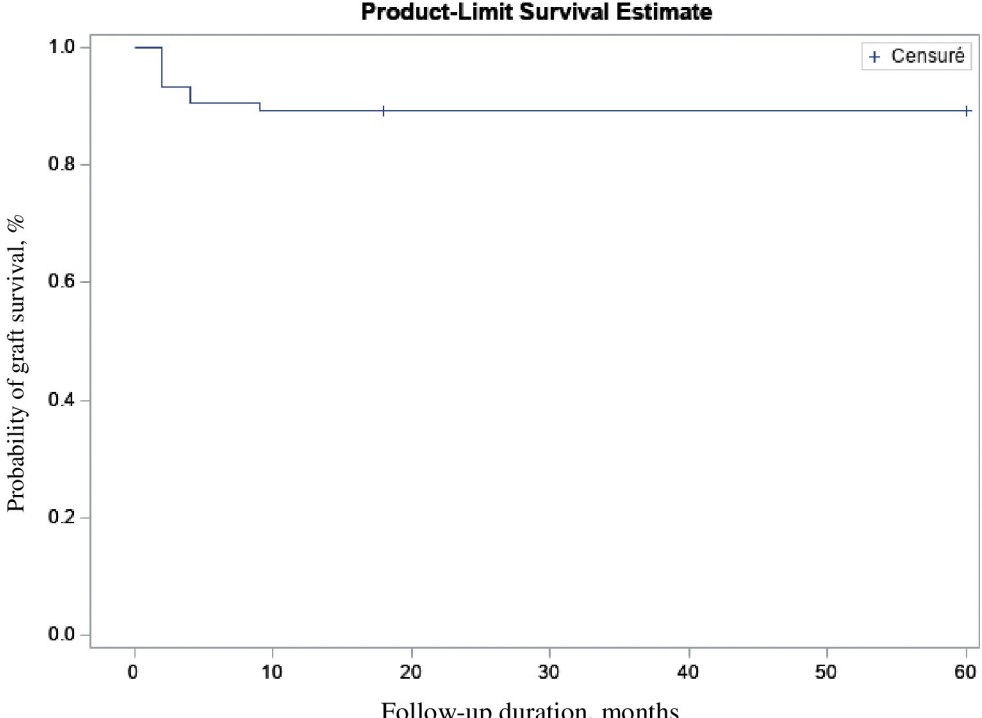

**Fig 2. Kaplan-Meier analysis of the DMEK graft survival probability of the whole cohort over 5 years (n = 107).**

postoperative timepoints relative to preoperative BCVA were statistically significant (all p<0.0001) (Table 3). After 5 years, 92% of eyes had a BCVA of ≥20/40 (≥0.5) while 77%, 57%, and 24% had BCVAs of ≥20/25 (≥0.8), ≥20/20 (≥1.0), and ≥20/17 (≥1.2), respectively (Fig 3B).

The median (IQR) graft ECD decreased sharply from 2550 (2433–2717) to 1350 (1026–1700) cells/mm$^2$ at 6 months (p<0.0001 relative to preoperative values): the median endothelial-cell loss (ECL) was 47%. Thereafter, the ECL continued at a lower rate (67 cells/year; 4.0% of the original ECD per year): by 5 years, the ECD was 900 (706–1200) cells/mm$^2$, which represents a total ECL of 65%. The changes at all postoperative timepoints relative to preoperative ECD were significant (all p<0.0001) (Table 3 and Fig 4).

The CCT values were frequently lacking. However, 97 of 107 preoperative values (91%) and 70 of 93 5-year values (75%) were available. These data showed a significant change in CCT from a median (IQR) of 618 (580–647) to 551 (520–578) μm. The changes at all postoperative timepoints relative to preoperative CCT were significant (all p<0.0001) (Table 3).

## Discussion

Our study is the ninth study on the long-term outcomes of DMEK (Table 1) [16–23]. The eight previous studies come from five groups, two of which sequentially published the 5- and 8-year (the Kruse group) [16,22] and 4–7-, 5-, and 10-year (the Melles group) [17–19] outcomes of their cohorts. The Melles cohort contains the first DMEK cases in the world and were conducted in 2007–2012 [17–19]. The Kruse-group cases were also conducted in this early period [16,22]. Thus, these cohorts may not necessarily reflect the clinical practices that have evolved since this period. Moreover, two of the three studies with more recent cases were

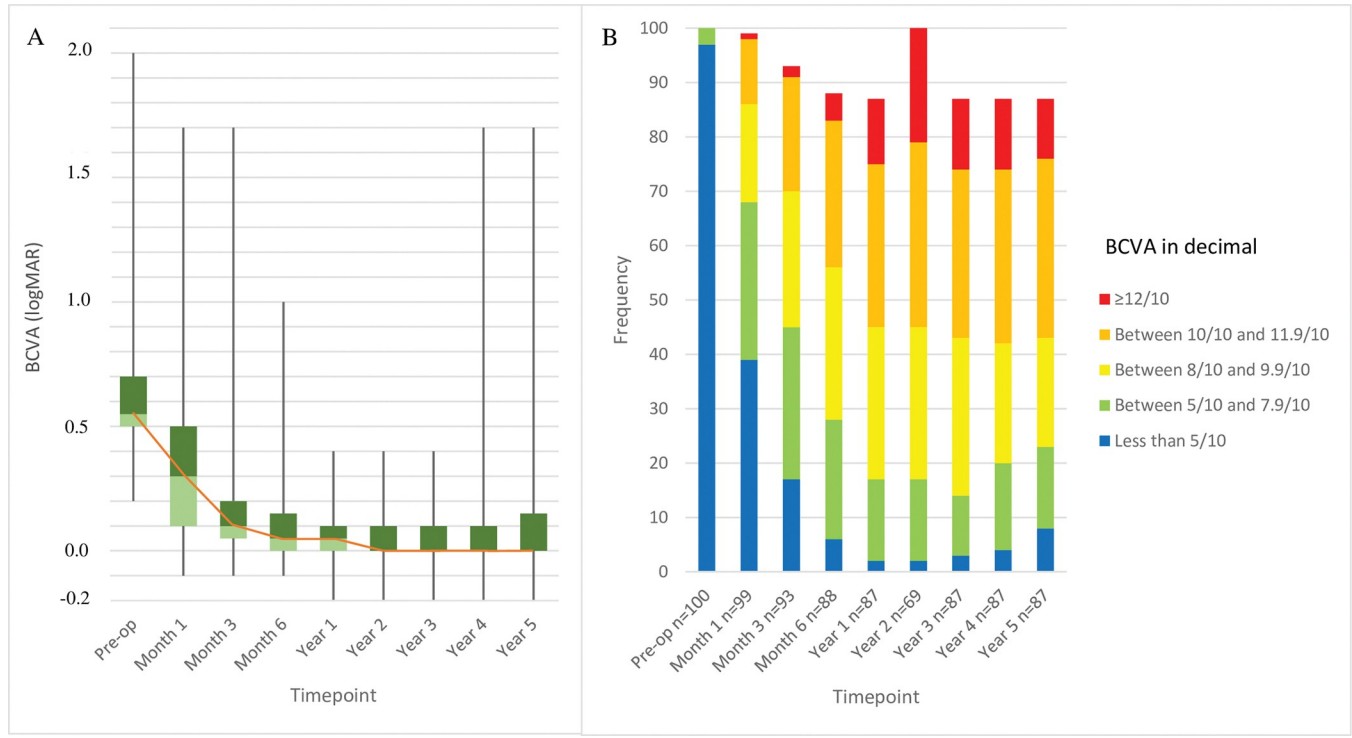

**Fig 3.** Average change over time in BCVA expressed as logMAR (A) and decimal categories (B) in the whole cohort (n = 100). Seven eyes with preoperative conditions that could affect visual acuity recovery were excluded from this analysis. (A) shows the median (orange line), interquartile range (green boxes), and maximum and minimum data as box-and-whisker plots.

limited by the fact that their cohorts were selected rather than being consecutive real-world case series: they excluded patients with extracorneal vision-limiting conditions from all analyses and focused on FECD and PBK patients only [21,23]. Moreover, the remaining study by Wardeh showed only one postoperative datapoint, namely, the average outcomes at last follow-up, which ranged widely from 20 to 80 months [20]. Therefore, we examined the 5-year outcomes of our own real-world case series of unselected consecutive patients who underwent first-time DMEK in 2014–2018.

Our results are compared to the other eight studies below. In terms of cohort composition, the sample sizes ranged from 66 to 500 eyes (vs. 107 eyes in our study). Except for the study by Besek et al. that involved 59% PBK cases [21], FECD was the most common indication in all studies (89–94% vs. 94% in our study), followed by PBK (1–8% vs. 3%) and failed keratoplasty (3–9% vs. 3%). The last timepoint examined was 3, 2–7, 5, 7, and 10 years (vs. 5 years in our study). The number of eyes that remained in the studies dropped substantially over time (Fig 5A) and in fact another candidate study was not included in Table 1 because only 15 of 121 DMEK cases were left at 3 years [28]. As shown by comparing Fig 5A (total loss) and 5B (loss due to graft failure), much of this was due to loss to follow-up. In our study, only 1% of the original cohort was lost to follow-up at 5 years. The patients in the eight Table 1 studies were 63–74 years old (vs. 72 years in our study). While the other studies showed female predominance (51–60%), which reflects the trend in FECD [29], our patients were particularly likely to be female (67%). The graft donors also tended to be older in our study (77 vs. 53–70 years). Triple-DMEK ranged widely between studies (0.8–51% vs. 32% in our study) (Table 1). This may reflect growing confidence in the safety of triple-DMEK [30].

**Table 3. Change in BCVA, graft ECD, and CCT over 5 years after DMEK for the whole cohort (n = 107).**

| Variable | Preop | Month 1 | Month 3 | Month 6 | Year 1 | Year 2 | Year 3 | Year 4 | Year 5 |
|---|---|---|---|---|---|---|---|---|---|
| BCVA† | n = 100 | n = 99 | n = 93 | n = 88 | n = 87 | n = 87 | n = 87 | n = 87 | n = 87 |
| /10 | | | | | | | | | |
| <5 | 97 (97) | 39 (39) | 17 (18) | 6 (7) | 2 (2) | 2 (2) | 3 (3) | 4 (5) | 8 (9) |
| 5–7 | 3 (3) | 29 (29) | 28 (30) | 22 (25) | 15 (17) | 13 (15) | 11 (13) | 16 (18) | 15 (17) |
| 8–9 | 0 (0) | 18 (18) | 25 (27) | 28 (32) | 28 (32) | 24 (28) | 29 (33) | 22 (25) | 20 (23) |
| 10–11 | 0 (0) | 12 (12) | 21 (23) | 27 (31) | 30 (34) | 30 (34) | 31 (36) | 32 (37) | 33 (38) |
| ≥12 | 0 (0) | 1 (1) | 2 (2) | 5 (6) | 12 (14) | 18 (21) | 13 (15) | 13 (15) | 11 (13) |
| Decimal | 2.75 (2–3) | 5 (3–8) | 8 (6–9) | 9 (7–10) | 9 (8–10) | 10 (8–10) | 10 (8–10) | 10 (8–10) | 10 (7–10) |
| logMAR | 0.6 (0.5–0.7) | 0.3 (0.1–0.5)* | 0.1 (0.05–0.2)* | 0.05 (0–0.15)* | 0.05 (0–0.1)* | 0 (0–0.1)* | 0 (0–0.1)* | 0 (0–0.1)* | 0 (0–0.15)* |
| Graft ECD | n = 107 | - | - | n = 94 | n = 93 | n = 93 | n = 93 | n = 92 | n = 92 |
| Cells/mm² | 2550 (2433;2717) | - | - | 1350 (1026;1700)* | 1212 (923;1564)* | 1100 (850;1413)* | 1000 (811;1309)* | 1000 (744;1254)* | 900 (706;1200)* |
| ECL‡ | - | - | - | -47 (-60; -35) | -53 (-64; -40) | -57 (-68; -46) | -61 (-69; -50) | -62 (-71; -52) | -65 (-72; -54) |
| CCT | n = 97 | - | n = 12 | n = 21 | n = 35 | n = 44 | n = 58 | n = 66 | n = 70 |
| µm | 618 (580;647) | - | 541 (536;555)* | 540 (535;560)* | 536 (516;562)* | 535 (512;558)* | 542 (518;575)* | 549 (516;575)* | 551 (520;578)* |
| Change§ | - | - | -15 (-19;-13) | -15 (-18;-9) | -13 (-18;-9) | -14 (-18; 7) | -11 (-17;-6) | -11 (-16;-6) | -10 (-16 4) |

Data are shown as n (%) or median (interquartile range [IQR]).

* Significantly different relative to preoperative values, as determined by Wilcoxon signed rank test followed by Bonferroni correction (all p<0.0001).

† Seven eyes with preoperative conditions that could affect visual function recovery were excluded from the BCVA analysis.

‡ Endothelial cell loss relative to preoperative values, expressed as %.

§ Change in CCT relative to preoperative values, expressed as %. Each calculation was relative to the preoperative CCT of the eyes that were avilable at the timepoint being examined.

BCVA, best-corrected visual acuity; CCT, central corneal thickness, DMEK, Descemet-membrane endothelial keratoplasty; ECD, endothelial cell density; ECL, endothelial-cell loss.

## Overall graft survival trends

The overall 5-year DMEK-graft survival probability was 88% in our study. In the literature, 5-year graft-survival rates ranged from 83–96% (Table 1 and Fig 5B). The 10-year analysis of the Kruse group recorded 96% graft survival at 7 years but they noted a sudden increase in graft failures starting at 8 years (6% of the eyes that attended the 8-year follow-up) [22]. By contrast, the Melles group reported 96% survival at 10 years and did not observe a sharp drop at any timepoint [19]. The reason for this difference is not clear. While multiple variables can affect graft failure after DMEK, including patient and donor age, preoperative-lens status, surgical indication, graft storage time, and duration of topical steroid treatment [21,30–32], the Kruse-group cohort did not display marked differences from the other studies in terms of these variables.

Notably, Besek et al. recorded a drastic and escalating reduction in graft survival: at the last timepoint (7 years), graft survival was 58%. This may reflect the fact that unlike the other cohorts, 58% of this cohort consisted of PBK patients (vs. 1–8%) [21]. Fajardo et al. [33] also reported that FECD eyes display significantly better 1-year DMEK-graft survival than PBK eyes (88% vs. 67%). Thus, for FECD at least, DMEK grafts seem stable for at least 5 years. However, the very long-term outcomes must be confirmed.

## Postoperative complications

The most prominent complication in our series was graft detachment that exceeded a third of the graft surface or threatened the visual axis and required rebubbling (18% of the 107 eyes).

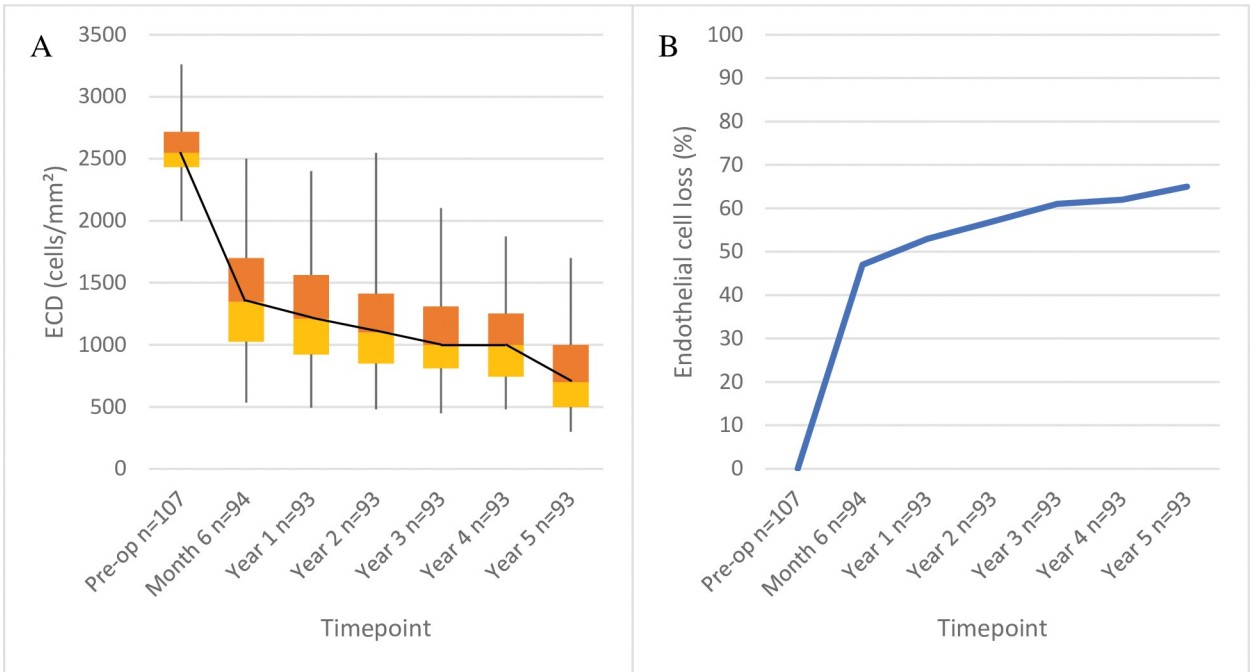

**Fig 4. Average change in graft ECD and endothelial cell loss over time in the whole cohort (n = 107).** (A) shows the median (black line), interquartile range (orange boxes), and maximum and minimum data as box-and-whisker plots. (B) shows median endothelial cell loss.

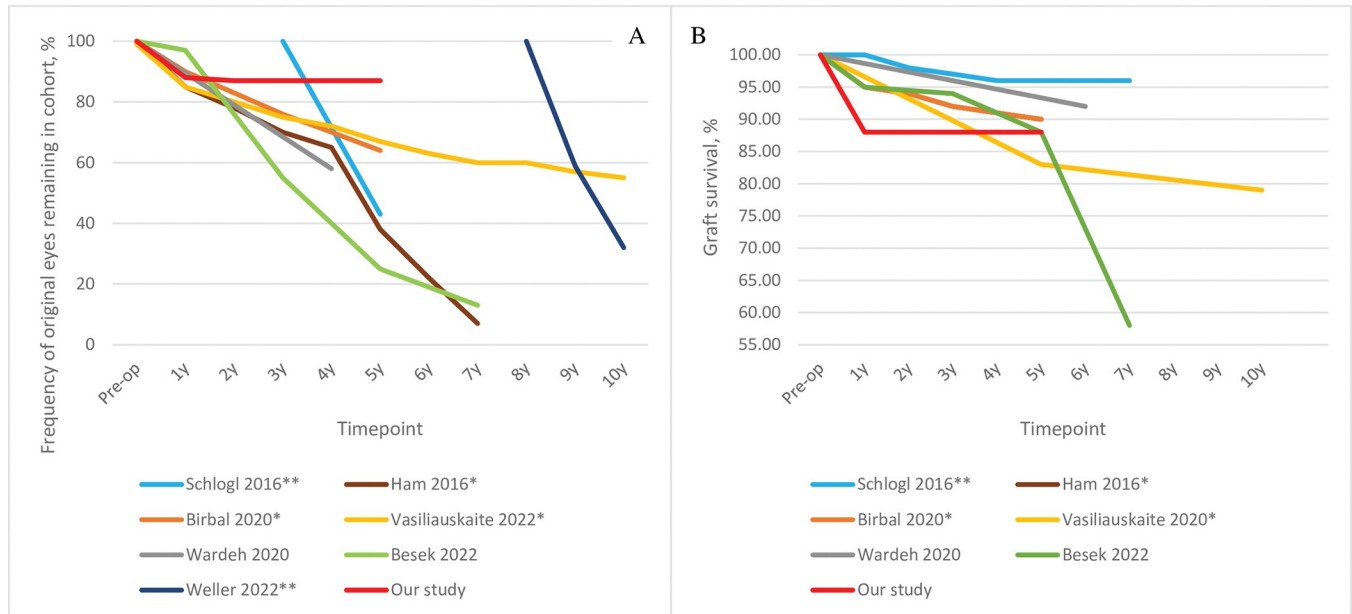

**Fig 5.** Comparison of our study (n = 107) with the literature in terms of (A) the percentage of original cohort eyes that did not attend follow-up and (B) graft survival. The study of Zwingelberg et al. [23] is not shown in (A) because only the patients who completed 3 years of follow-up were examined. The percentages in (A) are based on eye numbers in survival analyses [17–19] or the stated number of eyes at each follow-up [16,20–22]; thus, they include graft failures and loss to follow-up. * Studies on the Melles-group cohort; ** studies on the Kruse-group cohort.

Early graft detachment is a well-known Achilles heel of DMEK. It reflects the tendency of the graft to scroll tightly due to the elasticity of the Descemet membrane. This complicates graft generation from the donor cornea and placement of the graft against the host cornea, and reduces the ability of the bubble to promote graft adherence [34]. Major graft detachment was also the most common postoperative complication in the Table 1 studies, although its frequency ranged widely: it was 6–16% in the Melles group [17–19], 45% in the Kruse group [35], and 12–29% in the other groups [20,21,23]. Similarly, a recent systematic review by Deng et al. reported widely ranging graft-detachment rates after DMEK (0.2–76%; mean: 29%) [34]. This variation may partly reflect the use of rebubbling as a surrogate measure of graft detachment and the increasing use of rebubbling in the last decade.

In our study, rebubbling was successful in all but six cases (70%). Similarly, a recent prospective registry study by Dunker et al. showed that rebubbling was successful in 70% of DMEK eyes (19% rebubbling rate), and that graft-failure and rebubbling rates were inversely related [36].

Our overall graft failure rate was 12%. Notably, this rate ranged widely from 3% to 20% in the Table 1 studies. This reflects different study designs and sample sizes. Thus, although Besek et al. had a 20% graft failure rate, they also had a relatively small sample size (n = 150) and did not exclude any learning-curve eyes [37]. By contrast, Birbal et al. had a 3% graft failure rate but also had a large sample size (n = 500) and excluded the first 25 learning-curve eyes [38]. In our study, we decided to exclude only the worst of our learning curve (the first 5 eyes) so that our cohort represented a real-world cohort.

One of our SGF cases was due to allograft rejection (1%), which arose 15 months after DMEK. The Table 1 studies reported allograft rejection rates of 1–4.7% [16,18,19,21], while Deng et al. reported 1.9% (range: 0–5.9%) rates [34]. This confirms that this complication is rare in DMEK, even over long-term periods.

CME occurred in 6% of our series, which is similar to rates reported in the Table 1 studies (3–9%) [16,20,35] and elsewhere (7–16%) [39,40].

## Best-corrected visual acuity

Our study showed that DMEK yielded excellent visual-acuity outcomes (0.05 logMAR) a few months after surgery that were sustained for up to 5 years after DMEK. Similar outcomes were obtained by the Melles group (Table 1 and Fig 6) [18,19].

We observed that there was a sharp improvement in BCVA in the first 3 months and stabilization of BCVA at 1 year that persisted for up to 5 years post-DMEK. This was also observed by the other studies (Fig 6) [16,18,19,23,35]. Multiple shorter-term studies have also shown that BCVA improves during the first 3 [16] or 6 months [17,41] after DMEK and then stabilizes.

It should be noted that we expressed BCVA as median (IQR) because these data were not normally distributed (Fig 3A). All Table 1 studies expressed this variable as mean. Our mean preoperative and 5-year postoperative BCVAs were 0.6 and 0.1 logMAR, respectively.

## Graft endothelial-cell density

In our cohort, ECD dropped by 65% by 5 years post-surgery. The Melles and Kruse groups reported 5-year ECLs of 51–59% and 44–49%, respectively. The 10-year rates for these cohorts were 68–72% [16,18,19,22]. Our study and the other Table 1 studies showed that the vast majority of the ECL occurred in the first 6 months (Table 1 and Fig 7). This is consistent with many shorter-term studies [42–44] as well as a 5-year follow-up case-series study by Feng et al.

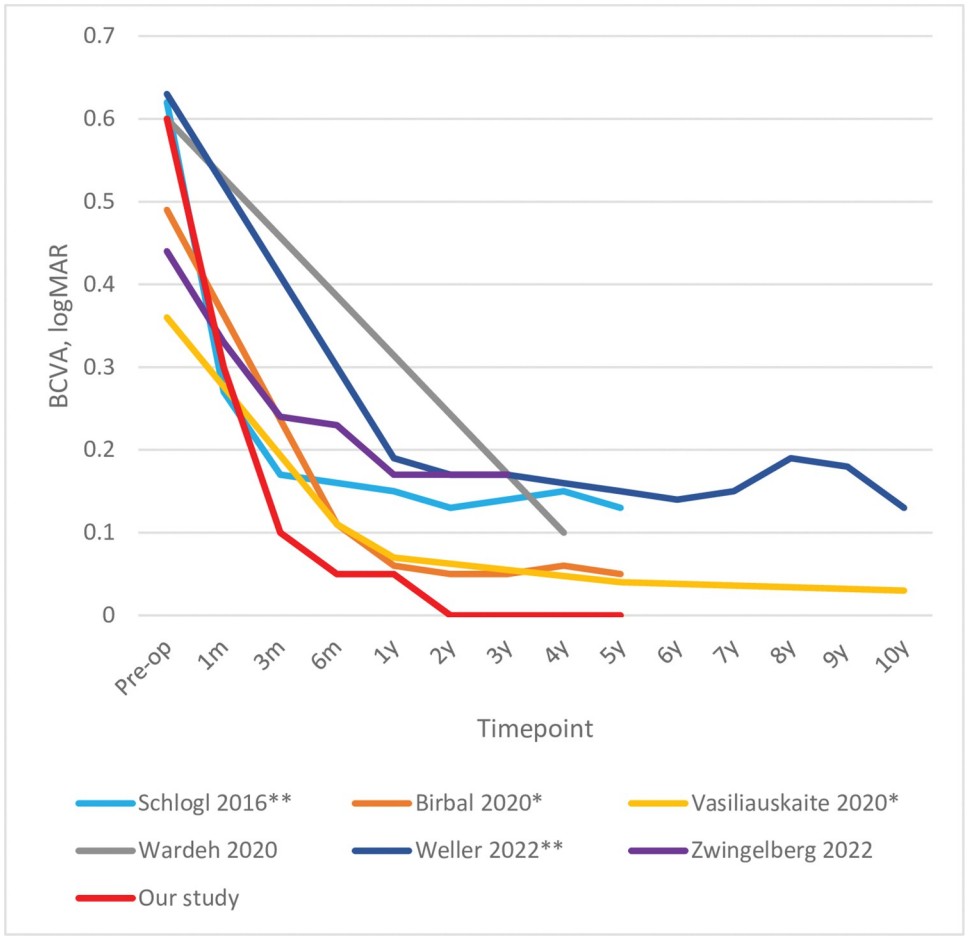

**Fig 6. Comparison of our study (n = 100) with the literature in terms of long-term visual outcomes of DMEK.** The percentage of the original cohort that was available at each timepoint is indicated. * studies on the Melles-group cohort; ** studies on the Kruse-group cohort. All visual acuity analyses excluded eyes with preoperative conditions that could interfere with visual acuity recovery.

that focused on ECD only (it was not included in Table 1 because only 28 of 962 eyes remained at 5 years).

Except for the Kruse-group studies, all other studies (and our own) observed ongoing linear ECD decreases ranging from 2.4%/year to 4.8%/year (4%/year in our study). Feng et al. also showed a 2.6% annual ECL over 5 years after stabilization. By contrast, the Kruse group observed only minor additional ECD losses after stabilization at 1 month (0.6–0.7%/year), but a sudden ECL of 28% was observed at year 7. The ECL then stabilized but was responsible for a surge of late graft failures [16,22]. Such a sharp drop in ECD between 7 and 10 years was not observed by the Melles group (Fig 7) [19]. The reason for these differences is not clear.

Notably, the average yearly ECL of the other Table 1 studies and our study after the initial postoperative drop was 3.7%, which is considerably higher than that reported for healthy eyes (0.3–0.6%). The reasons for this accelerated ECL after DMEK are unclear but may include transplantation-related factors that weaken the graft endothelium, including the manipulation during surgery, and/or corneal disease factors that create a hostile environment for the graft endothelial cells. The high ongoing ECL after DMEK confirms the vital importance of using grafts with a high ECD.

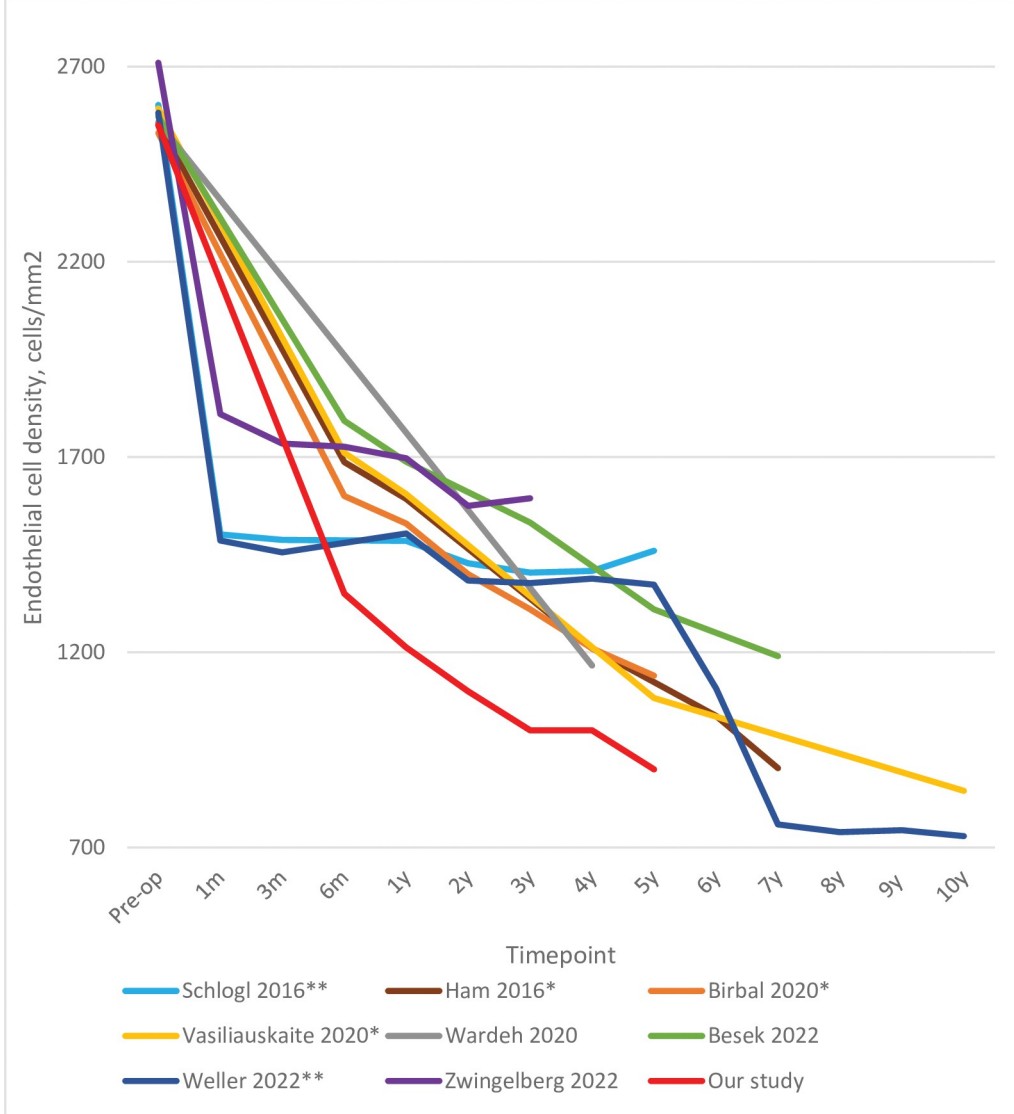

**Fig 7. Comparison of our study (n = 107) with the literature in terms of the long-term graft endothelial cell changes after DMEK.** The percentage of the original cohort that was available at each timepoint is indicated. * studies on the Melles-group cohort; ** studies on the Kruse-group cohort.

## Central-corneal thickness

In our study, only the preoperative and 5-year CCT values were sufficient for statistical comparison. This analysis showed that CCT dropped significantly by 67 μm. The Table 1 studies showed similar changes. Time-course analyses also showed that the CCT drop occurred immediately after surgery and then rose steadily over time by an annual average of 16 (range: 10–26; our study; 14) μm (Fig 8) [16–23].

## Study limitations and strengths

The retrospective design of our study is a limitation. The study also had a relatively small sample size (n = 107 eyes) compared to several of the other long-term DMEK studies (n = 230–500) [17,18,20,23]. However, our sample size was equivalent to or larger than that in the other

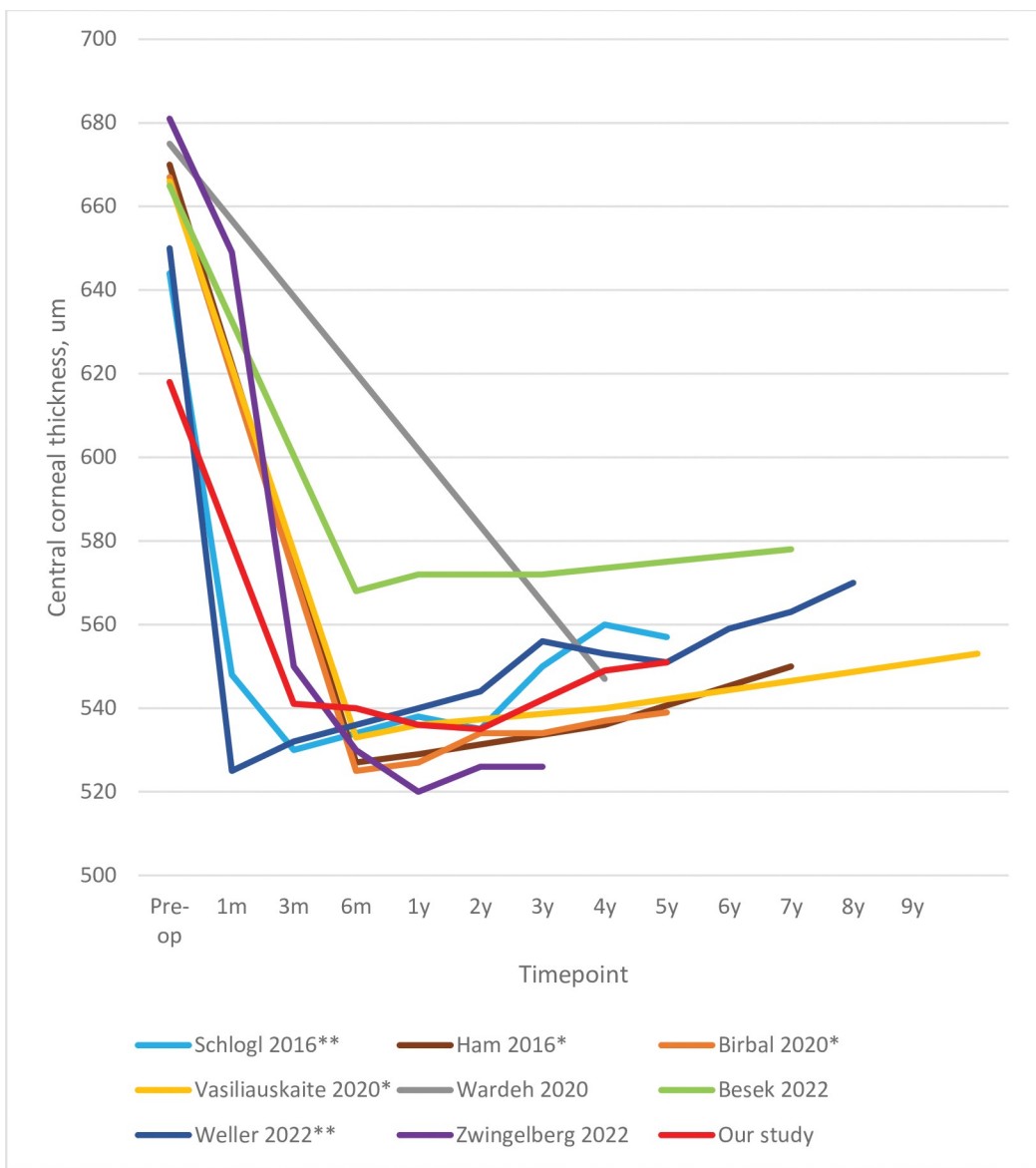

**Fig 8. Comparison of our study (n = 97) with the literature in terms of the long-term CCT changes after DMEK.** The percentage of the original cohort that was available at each timepoint is indicated. * studies on the Melles-group cohort; ** studies on the Kruse-group cohort.

studies (n = 66–150) [16,19,21,35]. Moreover, our study, like most of the other studies, largely examined FECD patients. However, there are also other indications for DMEK, especially PBK [7]. The long-term study with 59% PBK:41% FECD eyes by Besek et al. showed that PBK associated with worse graft survival and final BCVA than FECD [21]. By contrast, another long-term study (8% PBK:92% FECD eyes) showed no difference in final visual acuity, or ECD (survival was not assessed). However, the PBK eyes did exhibit consistently higher peripheral corneal thickness and although their CCT dropped to FECD eye levels at 1 year, they rose sharply thereafter, unlike in the FECD eyes [23]. Thus, while our findings are reflective of the real-world patient composition in DMEK, it remains to be determined whether similarly good long-term findings will be observed for other indications. Finally, a notable strength of our study was the low loss to follow-up, which was more severe in the other studies (Fig 5A).

## Conclusions

Our study together with the studies of five other groups show that the long-term clinical outcomes of DMEK are excellent and associate with low complication rates and overall high graft-survival probability (88% in our study at 5 years). Since DMEK is superior to DSAEK in terms of visual rehabilitation and has comparable ECL and graft survival, DMEK should be considered as the first-choice treatment, particularly for FECD. However, the durability of DMEK outcomes over longer periods and in PBK requires further examination.

## Supporting information

**S1 Fig.** Kaplan-Meier analysis of DMEK graft survival probability at 5 years for only the first eyes (n = 80) (A) and only the first-eyes of the FECD patients (n = 74) (B). DMEK, Descemet membrane endothelial keratoplasty; FECD, Fuchs endothelial corneal dystrophy.
(DOCX)

**S1 Table. Change in BCVA, ECD, and CCT over 5 years after DMEK for the first DMEK eyes (n = 80).**
(DOCX)

**S2 Table. Change in BCVA, ECD, and CCT over 5 years after DMEK for FECD first eyes (n = 74).**
(DOCX)

## Author Contributions

**Conceptualization:** Jean-Marc Perone.

**Data curation:** Pierre Bichet, Rémi Moskwa, Jean-Charles Vermion, Jean-Marc Perone.

**Formal analysis:** Christophe Goetz.

**Funding acquisition:** Christophe Goetz, Jean-Marc Perone.

**Investigation:** Pierre Bichet, Rémi Moskwa, Jean-Charles Vermion, Jean-Marc Perone.

**Methodology:** Christophe Goetz.

**Project administration:** Yinka Zevering, Jean-Marc Perone.

**Resources:** Christophe Goetz, Jean-Marc Perone.

**Software:** Christophe Goetz.

**Supervision:** Jean-Marc Perone.

**Validation:** Christophe Goetz, Jean-Marc Perone.

**Visualization:** Pierre Bichet, Yinka Zevering.

**Writing – original draft:** Pierre Bichet.

**Writing – review & editing:** Pierre Bichet, Rémi Moskwa, Christophe Goetz, Yinka Zevering, Jean-Charles Vermion, Jean-Marc Perone.

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
