## [Decision Letter · Decision Letter 0]

27 Sep 2023

PONE-D-23-17889Five-year clinical outcomes of 107 consecutive DMEK surgeriesPLOS ONE

Dear Dr. Perone,

Thank you for submitting your manuscript to PLOS ONE. After careful consideration, we feel that it has merit but does not fully meet PLOS ONE’s publication criteria as it currently stands. Therefore, we invite you to submit a revised version of the manuscript that addresses the points raised during the review process.

Please consider the comments from Reviewer #2 about the shortening of the manuscript to the relevant content. 

We look forward to receiving your revised manuscript.

Kind regards,

Timo Eppig

Academic Editor

PLOS ONE

Journal Requirements:

Reviewers' comments:

Reviewer's Responses to Questions

**Comments to the Author**

1. Is the manuscript technically sound, and do the data support the conclusions?

Reviewer #1: Yes

Reviewer #2: Yes

2. Has the statistical analysis been performed appropriately and rigorously? 

Reviewer #1: Yes

Reviewer #2: Yes

3. Have the authors made all data underlying the findings in their manuscript fully available?

Reviewer #1: Yes

Reviewer #2: Yes

4. Is the manuscript presented in an intelligible fashion and written in standard English?

Reviewer #1: Yes

Reviewer #2: Yes

5. Review Comments to the Author

Reviewer #1: It is good to have longer follow up data. This article also contains a comprehensive overview of the current lit.

1) Graft failure rate is very high. Were the PGFs evenly distributed betwen 2014 and 2018 or is there accumulation within the first 2 years? Could this be related to a learning curve beyond the first 5 patients? 5 patients is a low number for a DMEK learning curve, 20 or even more seems more realistic.

2) Please characterize the SGFs in more detail in the manusript (comparable to fig1): appearance min/max/average time after surgery

3) Standarddeviations are missing for some variables in the manuscript like BSCVA, CCT,... . Please add.

4) Table 3: equals baseline the preoperative measurements? Please clarify. Yeat 1-4 cells/mm2 were not significantyl different from baseline. If baseline is the preoperative donor cell density this should be significantly different.

5) Fig. 2 and M&M describe in more detail your definition for graft survival. How was edema in the diagnosis of graft failure defined? Was the timepoint of diagnosis of graft failure or timepoint of regraft chosen?

6) Fig 3a: SD is missing

7) Fig 3b: The legend and the graph ist self is not easy to understand. blue means <5/10, yellow >8/10 ,... decimal visual acuity. Please make legend more clear.

8) SD is missing

9) BCVA is extremely good. Do you have an explanation for that.

Reviewer #2: This is an interesting and important manuscript on “Five-year clinical outcome of 107 consecutive DMEK surgeries” from Mercy Hospital in Metz, France. Although the topic is not new and there are publications with far more patients, the comparison with already published data in a table and in well depicted figures makes this manuscript important.

However, the entire text can be shortened considerably. From my point of view, there is no need for 81 references. This may be shortened by half. Especially the entire DSAEK-topic can be removed from this manuscript to a major extent. In addition, there is no need for a discussion of 7 pages. This can be shortened by half. Also, what the author called “the Weller-group” is indeed the “Kruse-group” since Prof. Dr. Friedrich E. Kruse is the head of the Department of Ophthalmology in Erlangen, Germany and he was the first to popularize DMEK in Germany. For this reason, he should get the credit for his group throughout the manuscript.

Otherwise, here are some minor details regarding the manuscript:

Line 40: “Thirteen grafts failed” – please add the percentage

Line 77: “… indications with DMEK.” Here, the authors may want to add the brand-new publication of Flockerzi et al. in the British Journal of Ophthalmology detailing that in Germany around 98% of all deep anterior lamellar keratoplasties are indeed DMEK and not DSAEK.

Flockerzi E et al. Descemet's membrane endothelial keratoplasty is the predominant keratoplasty procedure in Germany since 2016: a report of the DOG-section cornea and its keratoplasty registry. Br J Ophthalmol. 2023 Aug 16:bjo-2022-323162. doi: 10.1136/bjo-2022-323162. Epub ahead of print.

Line 94: Something is wrong with the grammar of this sentence. Please revise.

Lines 146/147: Please add the percentage for “sterile air or sulfur hexafluoride bubble”

Line 211: Please correct to “… the analyses were repeated”

Line 311: It must be stated that graft survival also depends on the length of the topical steroid application. In this context, authors may want to add their regimen for application of which type of steroids after DMEK in their department.

Line 491: “Fuchs” with capital “F” – Prof. Dr. Ernst Fuchs, Vienna around 1900

Line 554: Literature no. 34, Cursiefen et al. – The German name is missing

Line 761: What does the abbreviation “IQR” stand for?

Line 782: “… that did not attend…” please delete “were”

Figure 2: There should be adequate English descriptions for the x-axis and the y-axis. Also, the x-axis is missing units.

6. PLOS authors have the option to publish the peer review history of their article (what does this mean?). If published, this will include your full peer review and any attached files.

Reviewer #1: No

Reviewer #2: No

---

## [Author Response · Author response to Decision Letter 0]

8 Nov 2023

Point-by-point Response to Reviewer Comments

Editor Comments

Reply: We have uploaded a rebuttal letter, a revised manuscript with Track Changes, and a revised manuscript without tracked changes.

Please note that during revision, we realized we had shown first-eye data (n=80) in Figures 2–4 in the original manuscript rather than whole-cohort data (n=107). These figures have thus been revised to show whole-cohort data. There are no substantial differences, as is also indicated by Supplementary Table S1. Some minor changes to the figure legends have also been made because of this.

We also failed to add Supplementary Tables S1 and S2 to the original submission, even though they were cited in the Results section. These data have been added to the revised submission. We apologize for these errors.

Reviewer 1

It is good to have longer follow up data. This article also contains a comprehensive overview of the current lit.

Reply: Thank you very much for your thorough review of our manuscript. We have addressed all comments and feel our manuscript is much improved. 

Please note that during revision, we realized we had shown first-eye data (n=80) in Figures 2–4 in the original manuscript rather than whole-cohort data (n=107). These figures have thus been revised to show whole-cohort data. There are no substantial differences, as is also indicated by Supplementary Table S1. Some minor changes to the figure legends have also been made because of this.

We also failed to add Supplementary Tables S1 and S2 to the original submission, even though they were cited in the Results section. These data have been added to the revised submission. We apologize for these errors.

Point 1.

Graft failure rate is very high. Were the PGFs evenly distributed between 2014 and 2018 or is there accumulation within the first 2 years? Could this be related to a learning curve beyond the first 5 patients? 5 patients is a low number for a DMEK learning curve, 20 or even more seems more realistic.

Reply: This is a good point. Of the 13 graft failure cases, seven (54%) occurred in the first 40 patients (37% of the cohort); thus, it is likely that a learning curve may have influenced our graft-failure rates. However, we chose to exclude only the first 5 learning-curve patients as a compromise between showing real-world cohort outcomes and not letting the very early-learning curve give an unrepresentative overall picture of DMEK ourcomes. 

 To address this, we added the following text to the Discussion:

“Our overall graft failure rate was 12%. Notably, this rate ranged widely from 3% to 20% in the Table-1 studies. This reflects different study designs and sample sizes. Thus, although Besek et al. had a 20% graft failure rate, they also had a relatively small sample size (n=150) and did not exclude any learning-curve eyes [37]. By contrast, Birbal et al. had a 3% graft failure rate but also had a large sample size (n=500) and excluded the first 25 learning-curve eyes [38]. In our study, we decided to exclude only the worst of our learning curve (the first 5 eyes) so that our cohort represented a real-world cohort.” Lines 341–347

Point 2.

Please characterize the SGFs in more detail in the manusript (comparable to fig1): appearance min/max/average time after surgery

Reply: We changed the text as follows:

“Of these 13 failure cases, seven were PGF cases: one failed before 1 month and six failed between 1 and <3 months (overall median, 2 months). The remaining six were SGF cases: five failed between 3 and <6 months and one failed at 15 months (overall median, 5 months).” Lines 236–239

Point 3.

Standarddeviations are missing for some variables in the manuscript like BSCVA, CCT,... . Please add.

Reply: Normality testing showed that BCVA, ECD, and CCT were generally not normally distributed, so the data are shown as median (interquartile range [IQR]) in Table 3 (rather than as mean±SD). To address this comment, we added the interquartile (IQR) ranges to the Results text, as follows:

“Thus, median (IQR) BCVA in the whole cohort improved steadily from 0.6 (0.5–0.7) to 0.05 (0–0.1) logMAR at year 1, after which it dropped further to 0 (0–0.1) logMAR at 2 years and stabilized (Figure 3A). At 5 years, BCVA had decreased by 0.6 logMAR to 0 (0–0.15) logMAR. The changes at all postoperative timepoints relative to preoperative BCVA were statistically significant (all p<0.0001) (Table 3).” Lines 258–262

“The median (IQR) graft ECD decreased sharply from 2550 (2433–2717) to 1350 (1026–1700) cells/mm2 at 6 months (p<0.0001 relative to preoperative values): the median endothelial-cell loss (ECL) was 47%. Thereafter, the ECL continued at a lower rate (67 cells/year; 4.0% of the original ECD per year): by 5 years, the ECD was 900 (706–1200) cells/mm2, which represents a total ECL of 65%. The changes at all postoperative timepoints relative to preoperative ECD were significant (all p<0.0001) (Table 3 and Figure 4).” Lines 265–270

“These data showed a significant change in CCT from a median (IQR) of 618 (580–647) to 551 (520–578) µm. The changes at all postoperative timepoints relative to preoperative CCT were significant (all p<0.0001) (Table 3).” Lines 271–275

Point 4.

Table 3: equals baseline the preoperative measurements? Please clarify. Yeat 1-4 cells/mm2 were not significantyl different from baseline. If baseline is the preoperative donor cell density this should be significantly different.

Reply: Yes, baseline means preoperative. Indeed, all postoperative BCVA, ECD and CCT changes were significantly different relative to the preoperative values, including after Bonferroni correction. 

To address this comment, the term “baseline” was changed to “preoperative values” throughout the manuscript. For example, the Methods text was changed as follows:

“Postoperative change in the clinical variables relative to preoperative values was assessed for statistical significance with Wilcoxon signed rank test.” Lines 204–205

Moreover, asterisks indicating significant differences relative to preoperative values were added to Table 3 (and Supplementary Tables S1 and S2, which show the same data but for first eyes and FECD first eyes only, respectively). 

Point 5.

Fig. 2 and M&M: describe in more detail your definition for graft survival. How was edema in the diagnosis of graft failure defined? Was the timepoint of diagnosis of graft failure or timepoint of regraft chosen?

Reply: We indicated our follow-up visit timepoints, indicated how edema was defined by adding the method (anterior segment-optical coherence tomography) and a reference (Ref 27), and clarified our definition of graft survival, as follows:

“Follow-up with complete ophthalmological testing was conducted 1, 8, and 15 days and 1, 3, 6, and 12 months after surgery and every year thereafter. ... Primary-graft failure (PGF) was defined as the failure of the pre-existing edema to resolve within 3 months of surgery, as measured with anterior segment-optical coherence tomography [27] at follow-up visits. Secondary-graft failure (SGF) was defined as the new emergence of corneal edema after 3 months. All PGF and SGF cases were treated with regraft with DMEK or DSAEK.” Lines 158–176

Point 6.

Fig 3a: SD is missing

Reply: We changed the BCVA data in Fig 3A to those of the whole cohort (rather than the first eyes shown in the original Fig 3A) and displayed the median (IQR) data in a box-and-whisker plot. This resulted in some changes to the Fig 3A legend (Lines 641-642) 

Point 7.

Fig 3b: The legend and the graph ist self is not easy to understand. blue means <5/10, yellow >8/10 ,... decimal visual acuity. Please make legend more clear.

Reply: We changed the BCVA data in Fig 3B to those of the whole cohort (rather than first eyes in shown the original Fig 3B) and clarified the legend. 

Point 8.

SD is missing

Reply: We changed the ECD data in Fig 4 to those of the whole cohort (rather than first eyes in the original Fig 4). The median (IQR) ECD values are shown as a box-and-whisker plot. This resulted in some changes to the Fig 3A legend (Lines 644-645)

Point 9.

BCVA is extremely good. Do you have an explanation for that.

Reply: For the BCVA analyses, we excluded the 7 eyes with ocular comorbidities that would have affected visual recovery. We also used median to express the data rather than mean. If we had included the 7 eyes with comorbidities and expressed the BCVA data as mean, the preoperative and postoperative BCVAs would have been 0.7 and 0.1 logMAR, respectively (rather than 0.6 and 0 logMAR, respectively).

 To address this point, we added the following text to the Discussion:

“It should be noted that we expressed BCVA as median (IQR) because these data were not normally distributed (Fig. 3A). All Table-1 studies expressed this variable as mean. Our mean preoperative and 5-year postoperative BCVAs were 0.6 and 0.1 logMAR, respectively.” Lines 363–365

We also added a footnote to the literature summary table (Table 1):

“a BCVA, ECD, and CCT data are expressed as mean in the literature but as median in our study because these variables were generally not normally distributed. If we had expressed these variables as means, preoperative BCVA (without 7 eyes with ocular comorbidities), ECD, and CCT would be 0.7 logMAR, 2574 cells/mm2, and 623 µm, respectively; and 5-year postoperative BCVA, ECD, and CCT would be 0.1 logMAR, 900 cells/mm2, and 561 µm, respectively.” Lines 580-583

Reviewer 2

This is an interesting and important manuscript on “Five-year clinical outcome of 107 consecutive DMEK surgeries” from Mercy Hospital in Metz, France. Although the topic is not new and there are publications with far more patients, the comparison with already published data in a table and in well depicted figures makes this manuscript important.

Reply: Thank you very much for your careful and through review. We have addressed all comments and feel that our manuscript has improved greatly.

Please note that during revision, we realized we had shown first-eye data (n=80) in Figures 2–4 in the original manuscript rather than whole-cohort data (n=107). These figures have thus been revised to show these data. There are no substantial differences, as is also indicated by Supplementary Table S1. Some minor changes to the figure legends have also been made because of this.

Point 1.

However, the entire text can be shortened considerably. From my point of view, there is no need for 81 references. This may be shortened by half. Especially the entire DSAEK-topic can be removed from this manuscript to a major extent. 

Reply: We have removed all mention of DSAEK (and PKP) from the Discussion as well as their associated references. The discussion of DSAEK was also made less prominent in the Introduction. As a result of these changes, the reference number has dropped from 81 to 47 and the total page number from 20½ to 18½. 

Point 2.

In addition, there is no need for a discussion of 7 pages. This can be shortened by half. Also, what the author called “the Weller-group” is indeed the “Kruse-group” since Prof. Dr. Friedrich E. Kruse is the head of the Department of Ophthalmology in Erlangen, Germany and he was the first to popularize DMEK in Germany. For this reason, he should get the credit for his group throughout the manuscript.

Reply: We agree that Prof. Kruse has contributed enormously to the development of DMEK and sincerely apologize for our error. We have renamed “the Weller group” as “the Kruse group” throughout the manuscript and figures.

 The Discussion was reduced to 5 pages by deleting the texts on DSAEK and PKP.

Point 3.

Otherwise, here are some minor details regarding the manuscript:

Line 40: “Thirteen grafts failed” – please add the percentage

Reply: The percentage (12%) has been added. Line 38

Line 77: “… indications with DMEK.” Here, the authors may want to add the brand-new publication of Flockerzi et al. in the British Journal of Ophthalmology detailing that in Germany around 98% of all deep anterior lamellar keratoplasties are indeed DMEK and not DSAEK.

Flockerzi E et al. Descemet's membrane endothelial keratoplasty is the predominant keratoplasty procedure in Germany since 2016: a report of the DOG-section cornea and its keratoplasty registry. Br J Ophthalmol. 2023 Aug 16:bjo-2022-323162. doi: 10.1136/bjo-2022-323162. Epub ahead of print.

Reply: We added this reference (Ref 13). Line 75

Line 94: Something is wrong with the grammar of this sentence. Please revise.

Reply: We corrected it as follows:

“Thus, we report the outcomes of all consecutive eyes that underwent first-time DMEK in 2014–2018 in our institution. The cohort was not selected in any way. All eyes were examined. Separate analyses were also conducted with first eyes only. Unlike most other long-term follow-up studies (Table 1), the vast majority of the patients were available at the 5-year follow-up.” Lines 90–94

Lines 146/147: Please add the percentage for “sterile air or sulfur hexafluoride bubble”

Reply: We use 20% SF6. This detail was added. Line 142

Line 211: Please correct to “… the analyses were repeated”

Reply: This correction was made. Line 208

Line 311: It must be stated that graft survival also depends on the length of the topical steroid application. In this context, authors may want to add their regimen for application of which type of steroids after DMEK in their department.

Reply: We added “duration of topical steroid treatment”. Line 315

 The steroid-treatment regimen that we use was detailed in the Methods as follows:

“After surgery, all patients were treated for 4 weeks with Maxidrol, which is a topical antibio-corticosteroid (Dexamethasone + Neomycin Polymyxine B; ALCON, Rueil Malmaison, France; four times/day), and an ophthalmic ointment (Vitamin A dulcis; ALLERGAN, Courbevoie, France; twice/day). Maxidrol was then tapered for a month and replaced with long-term low-dose corticosteroid eye drops (FLUCON; Novartis Pharma, Rueil Malmaison, France; three times/day).” Lines 152–157

Line 491: “Fuchs” with capital “F” – Prof. Dr. Ernst Fuchs, Vienna around 1900

Reply: The reference was actually incorrect (Ham et al. 2009) and has been corrected to Ham et al. 2016 (Ref #17) (Lines 470–473). The name Fuchs is not in the title of Ham et al. 2017. The other references were also checked to ensure they are the right ones.

Line 554: Literature no. 34, Cursiefen et al. – The German name is missing

Reply: We added the publisher details to this book. The reference (now Ref #29) now reads as follows:

“Cursiefen C, Jun A. Current Treatment Options for Fuchs Endothelial Dystrophy. Current Treatment Options for Fuchs Endothelial Dystrophy. 2016. Springer International Publishing. doi:10.1007/978-3-319-43021-8” Lines 510–512

Line 761: What does the abbreviation “IQR” stand for?

Reply: It stands for Interquartile Range. The abbreviation was spelled out. Lines 196, 592, and 624 

Line 782: “… that did not attend…” please delete “were”

Reply: ‘were’ was deleted. Line 647

Figure 2: There should be adequate English descriptions for the x-axis and the y-axis. Also, the x-axis is missing units.

Reply: We have added better y-axis (“Follow-up duration, months”) and x-axis (“Probability of graft survival, %”) titles to Figure 2.

---

## [Decision Letter · Decision Letter 1]

22 Nov 2023

Five-year clinical outcomes of 107 consecutive DMEK surgeries

PONE-D-23-17889R1

Dear Dr. Perone,

We’re pleased to inform you that your manuscript has been judged scientifically suitable for publication and will be formally accepted for publication once it meets all outstanding technical requirements.

Kind regards,

Timo Eppig

Academic Editor

PLOS ONE

Additional Editor Comments (optional):

Reviewers' comments:

Reviewer's Responses to Questions

**Comments to the Author**

1. If the authors have adequately addressed your comments raised in a previous round of review and you feel that this manuscript is now acceptable for publication, you may indicate that here to bypass the “Comments to the Author” section, enter your conflict of interest statement in the “Confidential to Editor” section, and submit your "Accept" recommendation.

Reviewer #2: All comments have been addressed

2. Is the manuscript technically sound, and do the data support the conclusions?

Reviewer #2: Yes

3. Has the statistical analysis been performed appropriately and rigorously? 

Reviewer #2: Yes

4. Have the authors made all data underlying the findings in their manuscript fully available?

Reviewer #2: Yes

5. Is the manuscript presented in an intelligible fashion and written in standard English?

Reviewer #2: Yes

6. Review Comments to the Author

Reviewer #2: (No Response)

7. PLOS authors have the option to publish the peer review history of their article (what does this mean?). If published, this will include your full peer review and any attached files.

Reviewer #2: No

---

## [Editor Report · Acceptance letter]

12 Dec 2023

PONE-D-23-17889R1 

Five-year clinical outcomes of 107 consecutive DMEK surgeries 

Dear Dr. Perone:

I'm pleased to inform you that your manuscript has been deemed suitable for publication in PLOS ONE. Congratulations! Your manuscript is now with our production department. 

Kind regards, 

on behalf of

Prof. Dr. Timo Eppig 

Academic Editor

PLOS ONE